# IF-MDM:
# Implicit Face Motion Diffusion Model for Compressing Dynamic Motion Latent

## Abstract

This paper introduces an implicit face motion diffusion model (IF-MDM), a fully self-supervised framework for learning dynamic facial motion tailored for audio-driven talking head generation. IF-MDM eliminates the need for explicit human head priors by utilizing implicit motion templates, effectively addressing common visual alignment issues between the head and the background, as well as the computational challenges associated with conventional, heavy latent diffusion-based methods. To enhance speech-motion alignment, our approach incorporates (1) local flow modules for fine-grained motion modeling, (2) motion statistics guidance to manage head pose and facial expression intensity, and (3) framewise temporal guidance to accurately capture phoneme-level dependencies in lip movements. IF-MDM achieves real-time performance, generating realistic and high-fidelity 512x512 resolution videos at up to 45 fps. By capturing subtle dynamic motions such as eye blinking and torso movements purely through self-supervised learning, our model extends its applicability beyond human faces, offering generalizable talking head generation for various characters and animals. For more details on this work, including supplementary materials and code, please visit our project page (`ifmdm.github.io`).

## 1 Introduction

Talking head generation synthesizes video sequences from an input facial image and corresponding speech audio. Given the inherently sparse information in speech signals, generating realistic videos is an ill-posed task necessitating sophisticated generative modeling techniques such as variational autoencoders (VAE) (Kingma, 2014), generative adversarial networks (GAN) (Goodfellow et al., 2014), and diffusion models (Ho et al., 2020; Rombach et al., 2022), alongside explicit facial representation methods including facial landmarks (Wang et al., 2019; Wu et al., 2021) and 3D morphable models (3DMM) (Gerig et al., 2018; Guo et al., 2020). Recent advancements in diffusion-based video generation have significantly improved visual quality and temporal coherence (Shen et al., 2023; Stypułkowski et al., 2024; Cui et al., 2025; Guo et al., 2024; Li et al., 2024; Wei et al., 2024). However, the substantial computational overhead and slow inference speed of these approaches impede their practical use, particularly for real-time applications and extended video sequences (Wei et al., 2024; Cui et al., 2025). Conversely, methods leveraging explicit facial priors offer computational efficiency and structured representations but suffer from compromised realism due to being overly dependent on face rendering, frequently resulting in misalignments of regions such as eyes and torso. Moreover, these methods rely heavily on pre-aligned and cropped facial data, inherently restricting the diversity and realism of generated head motions.

To address this challenge, we introduce a fully self-supervised learning framework that leverages implicit motion while eliminating reliance on human expert models for extracting motion liveness from video datasets. This approach enables the generation of videos that not only exhibit motion styles consistent with ground-truth sequences but also appear highly natural to human observers. However, it increases the overall learning difficulty. We mitigate this issue by constructing a motion latent space tailored for dynamic motion in talking head generation through the use of a Local Flow Module in conjunction with dataset curation and filtering. The Local Flow Module effectively captures global motion styles while simultaneously modeling fine-grained motions in facial sub-regions such as the

eyes and lips. Furthermore, structuring the dataset into progressive levels and gradually training the model proves highly beneficial for learning these multi-scale motion dynamics.

Beyond the challenge of disentanglement, implicit motion itself posed inherent difficulties. In particular, the dimensions learned in an unsupervised manner often appeared uncorrelated to the extent that they became impractical for diffusion models to learn. To address this, we introduced motion statistics as training cues, which not only stabilized learning but also provided motion controllability during inference. Another challenge stemmed from the weak correlation between speech and talking head motion, which we mitigated through architectural refinement via residual compression. Overall, by fully self-supervising the learning of implicit motion and removing human priors, we were able to extract motion liveness more directly from the dataset. However, we also observed that achieving controllability ultimately requires alignment mechanisms.

We conducted extensive experiments to analyze the strengths and limitations of the fully self-supervised compressed dynamic motion latent space. Quantitative and qualitative evaluations on HDTF (Zhang et al., 2021) and CelebV-Text (Yu et al., 2023) demonstrated that our approach outperforms existing state-of-the-art models, achieving superior visual quality and natural motion while generating 512×512 videos at 45 fps—surpassing video diffusion models in both efficiency and realism. Moreover, similar to studies employing 3D morphable models, we confirmed that the discovered motion statistics inherently provide motion controllability. An additional intriguing application of our framework lies in its applicability to animals, as it does not rely on human expert models; corresponding experiments are presented as well. Comprehensive ablation studies were also included to offer insights for future research.

## 2 RELATED WORKS

**Audio-driven talking head generation**  Early methods for audio-driven talking head generation used explicit facial priors like 3D morphable models or landmarks to animate facial regions (Prajwal et al., 2020; Thies et al., 2020; Fan et al., 2022). Generative models such as GANs and VAEs were later introduced to improve realism (Zhou et al., 2020). More recently, diffusion-based methods have achieved superior visual quality and temporal coherence (Shen et al., 2023; Wei et al., 2024; Cui et al., 2025), but their high computational cost limits real-time application. Our work addresses this limitation by using an efficient implicit motion representation optimized for diffusion-based generation, enabling scalable training without compromising fidelity.

**Motion transfer and implicit motion modeling**  Motion transfer aims to animate a source image with motion from a driving video. Techniques range from keypoint-based methods to implicit motion fields (Siarohin et al., 2019; Jeon et al., 2020). Recent studies have shown that decoder-aware implicit motion improves disentanglement (Wang et al., 2022). While prior work focuses on cross-identity reenactment, our goal is to learn a natural and expressive motion space specifically for speech, which requires fine-grained audio-visual modeling.

**Human Avatar**  Recent high-fidelity human avatar synthesis methods use NeRFs, 3D Gaussian Splatting, or volumetric capture to model speaker-specific geometry (Guo et al., 2021; Cho et al., 2024; Saito et al., 2024). Although these approaches produce excellent results, they often require extensive multi-view data and are not suitable for real-time applications. Industrial efforts like Codec Avatars also require specialized capture systems. In contrast, our lightweight, self-supervised framework synthesizes coherent motion from monocular video alone, offering broader applicability without identity-specific retraining.

**Comparison to Compressed Latent Diffusion Models**  The compressed latent diffusion paradigm has been adopted by models like VASA-1 (Xu et al., 2025), EMO (Tian et al., 2024), and GAIA (He et al., 2024), which show impressive progress. However, our key distinction is the use of a fully self-supervised approach that does not rely on human-specific priors. This allows our model to stably generate dynamic motions from academic-scale datasets. We acknowledge that direct quantitative comparisons are not feasible due to their closed-source nature, but qualitative results in our supplementary materials demonstrate comparable visual quality. Our self-supervised framework's ability to learn from video clips or multi-image inputs also enables talking head generation beyond human faces.

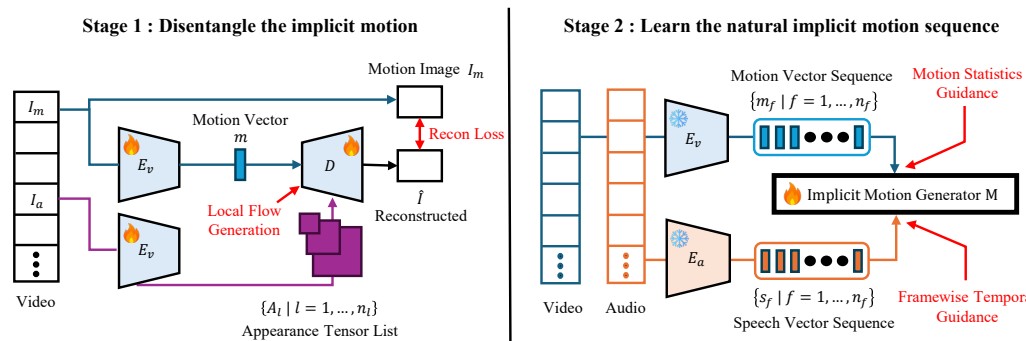

Figure 1: The training pipeline of our framework. In the stage 1 training the visual encoder $E_V$ and decoder $D$ is trained with the local flow module to disentangle the fine-grained motion vector $m$ and the appearance tensor list $\{A_l \mid l = 1, \ldots, n_l\}$. In the stage 2 training the diffusion-based implicit motion generator $M$ is trained with the extracted motion sequence $\{\hat{m}_f \mid f = 1, \ldots, n_f\}$ and speech vector sequence $\{s_f \mid f = 1, \ldots, n_f\}$ from the real video. The motion statistics guidance and the framewise temporal guidance is utilized on the stage 2.

## 3 METHODS

**Framework Overview**   The objective of our framework is to learn a compressed yet expressive implicit representation of dynamic facial and torso motions from video sequences, while simultaneously aligning these motions with corresponding speech audio. As depicted in Figure 1, the training pipeline is structured into two distinct stages: the first disentangles visual appearance from motion dynamics, and the second learns speech-conditioned temporal motion sequences.

During inference, as illustrated in Figure 2, the model synthesizes realistic talking head videos from an input identity image and speech audio. The appearance branch extracts identity-specific visual features, whereas the motion branch generates temporally coherent face motion sequences from the input speech. These two branches converge within the decoder, resulting in a synthesized video that integrates both appearance consistency and motion realism. A more detailed architecture can be found in Appendix B.

**Dataset Filtering**   We employed a two-stage dataset filtering process using facial landmarks and face segmentation maps to refine our training data. Initially, we processed 80,000 video clips to identify samples with distinct motion characteristics. We categorized these into three groups: a 'coarse motion' subset containing videos with only global head movement, a 'lip motion' subset focusing on videos with intricate lip movements, and a 'hard sample' subset that included videos with both types of movements. This filtering process reduced our dataset to 35,000 video clips, which were subsequently used to progressively train the Stage 1 model, ensuring it learned a robust representation for diverse motion types.

**Training Stage 1: Disentangle the implicit motion**   In the first stage, we employ a self-supervised approach using an encoder-decoder architecture to disentangle appearance and motion representations. Given an appearance image $I_a$ and a motion image $I_m$, both from the same video, the visual encoder $E_v$ extracts an appearance tensor list $\{A_l\}$ and a compact motion embedding vector $m$. The decoder $D$, employing a U-Net-like structure (Ronneberger et al., 2015) combined with progressive image synthesis (Karras et al., 2020), reconstructs a motion-transferred image $\hat{I}$ by warping appearance tensors with motion-conditioned flow fields.

Our objective is to capture authentic motion dynamics in a compressed latent form. To this end, we propose a **local flow module** that decomposes the global motion embedding into spatially localized flow fields, each designed to attend to semantically meaningful regions such as the lips, eyes, and torso.

Given the input feature map $\mathbf{F}_l$, we apply a grouped convolution with $G_l$ groups. The output is defined as:
$$\mathbf{F}_l^{\text{out}} = \text{GroupConv}(\mathbf{F}_l, G_l) \in \mathbb{R}^{B \times C_l \times H_l \times W_l},$$

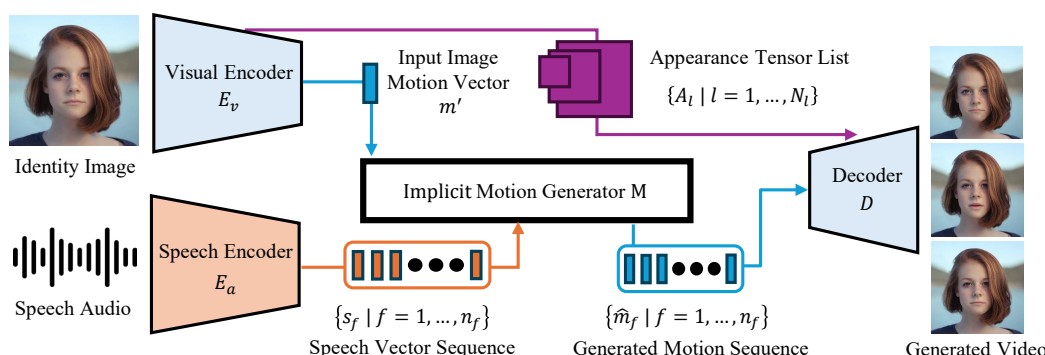

Figure 2: The inference pipeline of our framework. The visual encoder $E_v$ extract the motion vector of input image $m'$ and the appearance tensor list $\{A_l \mid l = 1, \ldots, n_l\}$ from the input image. The our diffusion-based implicit motion generator $M$ generate the motion sequence $\{\hat{m}_f \mid f = 1, \ldots, n_f\}$ by given speech vector sequence $\{s_f \mid f = 1, \ldots, n_f\}$ and the input image motion vector $m'$ as motion mean. Finally the decoder $D$ generate the video from the generated motion sequence and the appearance tensor list.

From $\mathbf{F}_l^{\text{out}}$, we generate a warping index $\mathbf{W}_l$ and a warping mask $\mathbf{M}_l$. To enforce spatial locality, we apply sequential 1D softmax operations along the height and width dimensions of the mask:

$$\mathbf{M}_l^{\text{norm}} = \text{Softmax}(\text{Softmax}(\mathbf{M}_l, \dim = -1), \dim = -2).$$

This normalization constrains each local flow to focus on compact, region-specific areas. By independently warping feature maps using $\mathbf{W}_l$ and $\mathbf{M}_l^{\text{norm}}$, the model synthesizes highly detailed and spatially coherent motion representations.

To facilitate stable training dynamics in the diffusion-based second stage, we apply motion space normalization using a `tanh` function. The local flow module compensates for the representational capacity lost from this normalization, enabling the learning of a motion embedding suitable for Stage 2 training.

**Training Stage 2: Audio-driven Motion Sequence Generation**   Our implicit motion generator $M$ is designed as a conditional denoising diffusion model to generate realistic motion embeddings aligned with speech and motion statistics. Given a noised motion sequence $\{m_f^t\}$ at diffusion timestep $t$, the model predicts the noise component $\epsilon_f^t$ for each frame:

$$\epsilon_f^t = M\left(\{m_f^t\}, \; t, \; \{s_f\}, \; m^\mu, \; m^\sigma, \; m'\right), \quad f = 1, \ldots, n_f, \tag{1}$$

where $m^\mu$ and $m^\sigma$ denote the motion mean and standard deviation, respectively, serving as *motion statistics guidance*. The speech vector sequence $\{s_f\}$ provides as *framewise temporal guidance*, encoding phoneme-level and prosodic structures extracted by the speech encoder. To enable long-term sequence generation, the model also incorporates the motion embedding of the previous frame $m'$ as a *motion hint*. These conditioning signals are injected into the diffusion transformer-like backbone via concatenation.

We empirically set the sequence length $n_f = 32$, which is sufficient to capture mid- to long-range facial dynamics. This design allows the model to synthesize temporally coherent and expressive talking head motions from implicit motion embeddings in a fully self-supervised setting.

To effectively guide the diffusion process, we introduce **motion statistics guidance** to capture global motion tendencies and intensity. Addressing the weak correlation between implicit motion and speech, we propose **framewise temporal guidance**. By integrating speech vector sequences directly into self-attention and modulation modules, we establish explicit frame-level temporal alignment, which markedly improves lip-sync quality.

**Inference**   At inference, the synthesized talking head video's motion characteristics can be explicitly controlled by adjusting the motion statistics parameters, $m^\mu$ and $m^\sigma$. Lower values of $m^\sigma$ yield

subtle and stable motions with enhanced lip-sync accuracy, whereas higher values allow for more expressive movements. This mechanism provides practical flexibility in adapting motion dynamics to diverse application requirements.

To enable long-term video generation, we utilize a recursive inference strategy. For the first chunk, the initial motion hint $m'$ is a learned, fixed motion vector. For all subsequent chunks, the motion hint is the last frame's generated motion embedding from the previous chunk. This approach ensures temporal continuity and coherence across long video sequences.

During inference, we additionally apply classifier-free guidance scaling (Ho & Salimans, 2021) exclusively to the speech condition to further enhance the alignment between speech and facial motion. Let $\epsilon_{\text{cond}}$ denote the noise prediction conditioned on all inputs and $\epsilon_{\text{uncond}}$ denote the prediction where only the speech vector is omitted. The final noise prediction used for denoising is then computed as:

$$\epsilon = \epsilon_{\text{uncond}} + w \cdot (\epsilon_{\text{cond}} - \epsilon_{\text{uncond}}), \tag{2}$$

where $w$ is the guidance scale, fixed to $2.5$. This selective guidance amplifies speech-conditioned features without perturbing the motion priors, thereby improving lip-sync fidelity and temporal consistency.

## 4 EXPERIMENTS

**Implementation Details**  We trained our model using the CelebV-Text (Yu et al., 2023) and HDTF (Zhang et al., 2021) datasets. The Wav2Vec model (Baevski et al., 2020) was employed to extract speech vector sequences, subsequently used for training the implicit motion generator $M$. As discussed in our analysis, these vectors capture phoneme-level and prosodic features essential for our framewise temporal guidance. Stage 1 training required approximately 10 days on four NVIDIA A6000 GPUs, whereas Stage 2 training took around one week using a single NVIDIA A100 80G GPU. We extensively referenced the codebases of LIA (Wang et al., 2022) and DiT (Peebles & Xie, 2023), implemented training and inference with the PyTorch framework (Paszke et al., 2017).

**Experiment Setup**  Consistent with previous studies, we allocated 30 videos from the HDTF dataset (Zhang et al., 2021) as our test set for quantitative evaluation. To evaluate our method's strengths comprehensively, we selected representative baseline methods spanning different paradigms: SadTalker (Zhang et al., 2023) and Real3DPortrait (Ye et al., 2024), which utilize explicit face models; AniPortrait (Wei et al., 2024) and Hallo3 (Cui et al., 2025), video diffusion-based approaches; and Ditto (Li et al., 2024), which also employs implicit motion representations. Additionally, we assessed performance on in-the-wild inputs using images licensed by Unsplash[1].

**Talking Head Generation**  As summarized in Table 1, our method consistently outperforms baseline methods across multiple metrics while achieving real-time performance exceeding 30 fps. This result directly supports our core hypothesis that an efficient implicit motion representation can surpass computationally intensive methods. Compared with video diffusion models such as AniPortrait and Hallo3 (Wei et al., 2024; Cui et al., 2025), IF-MDM delivers comparable visual quality (FID 42.84 vs. 49.13/48.78) and superior temporal consistency (VideoScore-TC 3.71 vs. 2.57/3.29), along with significantly faster generation speed (FPS 30.90 vs. 0.88/1.25). The lower temporal consistency in AniPortrait may result from its frame-by-frame restoration with GFPGAN (Wang et al., 2021), which disrupts coherence.

Our method's fully self-supervised approach addresses key limitations of explicit face models. Although Real3DPortrait (Ye et al., 2024) and SadTalker (Zhang et al., 2023) achieve strong lip-sync, their dependence on rendered face models introduces spatial inconsistency, creating unnatural "floating-head" artifacts (reflected by a poor FID score of 74.68). In contrast, our approach generates natural full-frame results, offering superior visual realism and comparable lip-sync accuracy (LSE-D 8.21 vs. 8.23), entirely without explicit facial priors.

---

[1] https://unsplash.com/license

Table 1: Quantitative results of talking head generation on the testset of HDTF dataset. The FPS is calculated on a NVIDIA A6000. The **bold** means the best score. The underline means the 2nd best score. The * means the model requires a H100 80GB for the inference.

| | Image Quality | Identity Preservation | Temporal Consistency | Lip-Sync | | Speed |
|---|---|---|---|---|---|---|
| | FID↓ | CSIM↑ | VideoScore-TC↑ | LSE-D↓ | LSE-C↑ | FPS↑ |
| **Explicit Face Model** | | | | | | |
| SadTalkerZhang et al. (2023)) | 54.58 | 0.981 | 3.21 | 8.66 | 6.06 | 8.86 |
| Real3DPortraitYe et al. (2024)) | 74.68 | 0.982 | 2.10 | 8.23 | 6.58 | 10.21 |
| **Video Diffusion Model** | | | | | | |
| AniPortraitWei et al. (2024) | 49.13 | 0.978 | 2.57 | 11.56 | 3.00 | 0.88 |
| Hallo3Cui et al. (2025)) | 48.78 | 0.976 | 2.97 | 9.73 | 3.98 | 1.25* |
| **Implicit Motion Model** | | | | | | |
| DittoLi et al. (2024)) | 51.46 | 0.983 | 2.70 | 10.74 | 3.91 | 4.07 |
| IF-MDM (ours) | **41.91** | **0.984** | **3.12** | **8.21** | **7.35** | **31.18** |
| Ground Truth | 0.00 | 1.00 | 2.71 | 8.48 | 6.28 | - |

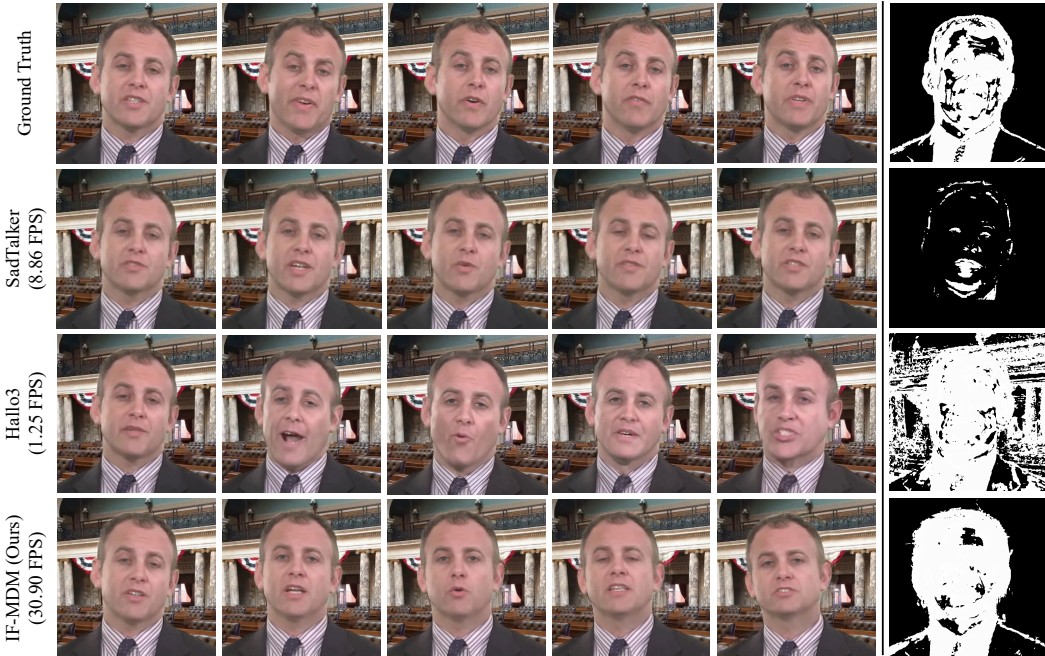

Figure 3: The qualitative results of HDTF datasets with baselines. Each column has same frame index on the reconstruction result with the first image and speech audio of ground truth. The rightmost column shows the merged difference from the first frame, visualizing the motion style. Please check the supplementary files for video results.

Compared to the implicit motion-based method Ditto (Li et al., 2024), our model demonstrates superior performance across all evaluation criteria, including lip-sync accuracy (LSE-C 7.35 vs. 3.91) and visual fidelity (FID 42.84 vs. 51.46). The slower performance of Ditto primarily arises from its reliance on post-processing for 3DMM extraction to enhance video quality. Our model, by comparison, achieves faster end-to-end inference, as illustrated in Figure 2. Although direct quantitative comparisons were limited by unavailable public code, we have included a supplementary video with reconstruction results on the project pages of VASA-1 (Xu et al., 2025), EMO (Tian et al., 2024), and GAIA (He et al., 2024) to enable a more comprehensive qualitative comparison.

**Local Flow Analysis** This analysis supports our claim that the local flow module enhances fine-grained motion modeling. We visually assess the module's impact, as illustrated in Figure 4. Incorporating local flow substantially improves motion coherence and preserves delicate facial details,

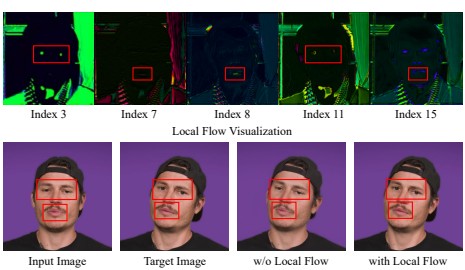

Figure 4: The visualization and comparison with and without local flow module.

| Modulation | Residual Addition for Self-Attention | LSE-D↓ |
|---|---|---|
| Global | X | 14.35 |
| Framewise Temporal | X | 11.11 |
| Framewise Temporal | O | **8.28** |

Table 2: Ablation Study for speech classifier-free guidance. The **bold** means the best score.

especially in dynamic areas like the mouth and eyes. Without this module, noticeable artifacts and degraded motion alignment occur.

The top row of Figure 4 displays localized flow fields, revealing that each index specializes in specific facial regions, such as the eyes and mouth. This specialization highlights the module's capacity to disentangle intricate motion components from global motion vectors, contributing to accurate motion representation in expressive areas. The bottom row compares results with and without local flow. Without it, overall head pose is maintained, but detailed motions (mouth and eyes) become static or misaligned, disrupting realism. In contrast, enabling local flow accurately captures local motions synchronized with speech, enhancing realism and temporal consistency. This analysis demonstrates that the local flow module effectively learns to capture fine-grained motion, rather than overfitting to only coarse motion across a larger field of view, as originally intended. Visualizations provide evidence for this, showing that the model concentrates on localized regions such as the eyes and mouth.

**Framewise Temporal Guidance** We evaluate framewise temporal guidance via ablation studies (Table 2). This analysis provides evidence for the effectiveness of our approach in improving speech-motion alignment, a key contribution. Replacing global modulation with framewise temporal modulation notably improves outcomes, reducing FID from 83.22 to 42.84 and LSE-D from 14.35 to 11.11, alongside increased CSIM from 0.95 to 0.97. The addition of FID and CSIM values to Table 2 provides a more complete picture of the performance gains. Furthermore, introducing residual addition in self-attention mechanisms provides additional gains, yielding the lowest FID (30.80), highest CSIM (0.98), and best lip-sync accuracy (LSE-D 8.28). This residual addition mechanism strengthens the direct temporal link between speech signals and motion embeddings at each self-attention layer, which is crucial for achieving high lip-sync accuracy. These findings underscore the efficacy of framewise temporal guidance combined with residual attention, significantly enhancing visual quality and speech-motion alignment.

Interestingly, similar experiments conducted with other modalities, such as text-to-image generation, do not exhibit such discrepancies. A key distinction lies in the nature of modality influence: while text typically induces global semantic transformations across the entire image, speech primarily induces localized effects on facial regions, with only limited global influence manifesting temporally—such as through prosody-driven head movements. This localized and weakly-correlated nature of speech-to-motion mapping likely imposes structural challenges for diffusion-based implicit motion modeling. Consequently, prior works often resorted to incorporating explicit geometric priors, such as 3DMM parameters, as classifier-free guidance signals to mitigate the ambiguity in the generation process. Our findings indicate that strong framewise modulation mechanisms are essential to overcome these modality-specific challenges and to establish reliable alignment between speech and motion representations.

**Ablation Studies** To further validate our design choices, we conducted ablation studies on two key components: the local-flow module and the dimensionality of the motion latent space.

First, we examined the effect of the local-flow module by varying the number of groups $G$. Without the module ($G = 1$), Stage 1 learning did not converge under our large field-of-view setting in Table 3. Similarly, $G = 4$ failed to converge, while $G = 8$ produced limited reconstruction performance (PSNR: 27.32 for same identity, 16.32 for different identity). Increasing to $G = 16$ yielded the

Table 3: Ablation Study for the number of group $G$ of local flow module.

| G | Same Identity | Diff Identity |
|---|---|---|
| 1 | - | - |
| 4 | - | - |
| 8 | **27.32** | **16.32** |
| 16 | 32.04 | 29.74 |
| 32 | 31.21 | 27.34 |

Table 4: Ablation study for dimension $D$ of motion vector.

| D | Same Identity | Diff Identity | Convergence on Stage 2 |
|---|---|---|---|
| 20 | 32.04 | 29.74 | YES |
| 40 | 33.01 | 26.32 | NO |
| 100 | 31.90 | 25.90 | NO |

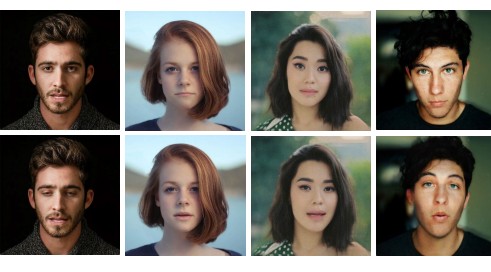

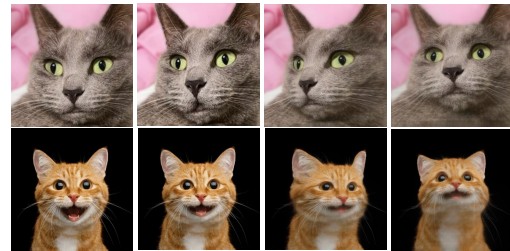

Figure 5: The qualitative results of the talking head generation with in the wild input image.

Figure 6: The scalability of our method is demonstrated by cat talking head generation.

best balance (32.04/29.74), while further increasing to $G = 32$ slightly degraded performance (31.21/27.34). The degradation with $G = 32$ likely results from the over-segmentation of the motion into too many small, non-meaningful groups, which hinders the learning of coherent, fine-grained motions. This confirms that the local-flow module not only enables convergence but also improves fine-grained motion learning in sub-parts such as eyes and lips.

Second, we analyzed the effect of motion latent dimensionality $D$ in Table 4. While a compact representation with $D = 20$ achieved strong performance (32.04/29.74), increasing to $D = 40$ slightly improved same-identity reconstruction (33.01) but reduced generalization across identities (26.32). Models with larger $D$ (e.g., 100) failed to converge in Stage 2 training, suggesting that excessive dimensionality impairs disentanglement between motion and appearance. These results highlight that compressed motion representations are not only efficient but also crucial for stable training and controllable inference.

To assess speed-quality trade-offs, we further varied diffusion steps (Table 5). Increasing steps generally improved generation quality (lower FID, higher CSIM) at the cost of inference speed. For instance, 100 diffusion steps balanced quality and efficiency (30.90 fps), while additional steps improved fidelity but reduced throughput, providing tunable performance depending on application needs.

**Towards Arbitrary Talking Head Generation** We further evaluated our model's generalization capabilities by testing it on in-the-wild images sourced from the internet. As depicted in Figure 5, IF-MDM consistently produces high-quality, coherent talking head videos across various real-world identities and challenging conditions. This outcome supports our claim that a fully self-supervised approach without human-specific priors offers broader applicability.

Additionally, Figure 6 demonstrates our model's capability to extend beyond human faces through personalization. Leveraging the fully self-supervised training approach that does not depend on human-specific expert models, IF-MDM facilitates arbitrary talking head generation. We initially trained the model on the AFHQ-Cat dataset (Choi et al., 2020) to capture diverse poses and expressions. Subsequently, using LoRA-based personalization (Hu et al., 2022), we successfully generated talking head outputs exhibiting varied poses, expressions, and mouth movements distinct from the original identity images. This highlights the model's robust adaptability to diverse and novel identity domains.

**Controlling Motion Characteristics** We explore motion controllability by varying the motion mean $m^\mu$ and motion standard deviation $m^\sigma$ (Table 6). We found that applying the motion mean from previous frames stabilizes pose and expression, while lower motion standard deviations enhance

Table 5: Ablation Study for the diffusion steps. The **bold** means the best score. The underline means the best trade-off for visual quality and the speed.

| Diffusion Steps | FPS↑ | FID↓ | CSIM↑ | LSE-D↓ |
|---|---|---|---|---|
| 50 | **45.75** | 50.22 | 0.981 | 11.11 |
| 100 | 30.90 | 42.35 | **0.984** | 10.65 |
| 200 | 20.22 | 31.85 | 0.982 | 10.63 |
| 500 | 9.90 | **30.80** | 0.981 | **10.56** |
| 1000 | 2.18 | 31.14 | 0.980 | 10.61 |

Table 6: Ablation study for motion mean and standard ($m^\mu$, $m^\sigma$). The **bold** means the best score.

| $m^\mu$ | $m^\sigma$ | FID↓ | CSIM↑ | VideoScore-TC↑ | LSE-D↓ |
|---|---|---|---|---|---|
| - | - | 46.21 | 0.981 | 2.67 | 11.46 |
| $m'$ | - | 42.84 | 0.983 | 2.54 | 12.79 |
| $\hat{m}^{n-1}$ | - | **28.83** | **0.984** | 2.99 | 11.05 |
| - | 0.1 | 42.84 | 0.983 | **3.42** | **8.41** |
| - | 0.3 | 58.66 | 0.973 | 2.40 | 9.78 |
| - | 0.6 | 73.27 | 0.970 | 2.47 | 10.77 |
| - | 0.9 | 103.22 | 0.968 | 2.51 | 11.26 |
| *GT* | *GT* | 20.88 | 0.984 | 2.79 | 8.77 |

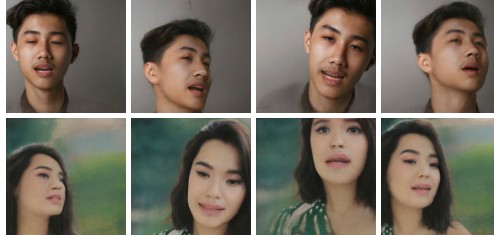

Figure 7: The effect of motion mean. (1st row) Without motion mean the generated video has diverse pose and motion. (2nd row) With motion mean the generated video shows stable pose and expression.

Figure 8: The effect of high motion standard value on inference. The generated video has dynamic motion, compromising the visual quality.

visual quality and lip-sync. Optimal results are obtained with $m^\mu = \hat{m}^{n-1}$ and $m^\sigma = 0.1$, achieving lowest FID (28.83), highest VideoScore-TC (3.42), and improved LSE-D (8.41), demonstrating effective alignment between generated motion and speech. A full exploration of the controllability space with different motion mean values (e.g., ground truth vs. randomly sampled) is left for future work, but our current findings confirm the practical utility of these parameters.

Qualitative analysis (Figures 7 and 8) confirms these findings. Without motion mean, motion becomes unstable and erratic, whereas applying $m^\mu$ produces consistent, realistic outputs. Similarly, increasing $m^\sigma$ exaggerates and destabilizes motion, highlighting its sensitivity in modulating realism and temporal coherence.

## 5 CONCLUSION

We introduced IF-MDM, a fully self-supervised implicit face motion diffusion model designed for real-time, high-fidelity talking head generation. Our two-stage training pipeline, which learns a compressed, decoder-aware motion representation through inter-frame reconstruction, enables efficient and expressive motion synthesis without reliance on explicit facial priors or 3D templates. By incorporating a novel local flow module, motion statistics guidance, and framewise temporal alignment, our model captures fine-grained expressiveness and achieves superior lip-sync accuracy. Through extensive experiments, IF-MDM demonstrates significant advantages over existing explicit and video diffusion-based approaches. It consistently outperforms baselines in visual quality, identity preservation, and, most notably, inference speed, producing $512\times512$ resolution videos at up to 45 fps. Its self-supervised formulation also enables successful generalization to diverse domains, including non-human characters and animals.

Despite its strengths, IF-MDM faces challenges in handling complex scenarios such as multi-person interactions and dynamic lighting. Future work will focus on improving expressiveness and robustness to these domain shifts. While the model has broad applicability in creative fields, we acknowledge the ethical risks associated with misuse in deepfakes and synthetic misinformation. We emphasize the importance of responsible deployment, urging the development of robust detection mechanisms and transparent usage practices to mitigate potential harms.

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

Table 7: The mean and standard deviation for results of our method in Table 1.

|  | Image Quality | Identity Preservation | Temporal Consistency | Lip-Sync | | Speed |
|---|---|---|---|---|---|---|
|  | FID↓ | CSIM↑ | VideoScore-TC↑ | LSE-D↓ | LSE-C↑ | FPS↑ |
| Mean | 37.12 | 0.985 | 3.22 | 9.31 | 7.33 | 31.90 |
| Std | 5.26 | 0.006 | 0.71 | 2.40 | 1.06 | 2.51 |

Table 8: Quantitative results of talking head generation on the testset of CelebV-Text dataset.

|  | FID | FVD | LSD-C | LSD-D |
|---|---|---|---|---|
| SadTalker | 50.015 | 471.163 | 6.922 | 7.921 |
| DreamTalker | 109.011 | 988.539 | 5.709 | 8.743 |
| AniPortrait | 46.915 | 477.179 | 2.853 | 11.709 |
| Hallo | 44.578 | 377.117 | 7.191 | 7.984 |
| Hallo3 | 43.271 | 355.272 | 6.527 | 9.113 |
| IF-MDM (Ours) | 43.252 | 301.119 | 7.252 | 8.351 |
| Ground Truth | 0.00 | 0.00 | 7.372 | 7.518 |

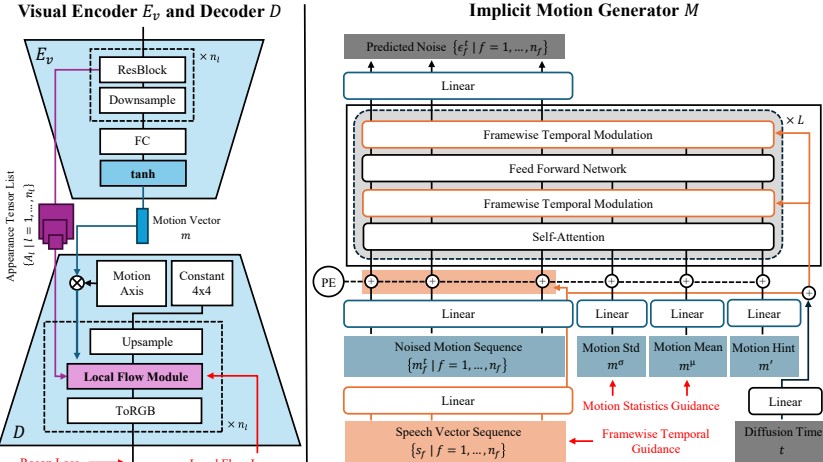

Figure 9: The detailed architecture of the visual encoder $E_v$, the decoder $G$, and the implicit motion generator $M$.

# A   MORE RESULTS ON THE TALKING HEAD GENERATION

In Table 7, we report the mean and standard deviation of key evaluation metrics derived from three independent experimental runs to ensure robust statistical validity. Specifically, metrics such as image quality (FID), identity preservation (CSIM), temporal consistency (VideoScore-TC), lip synchronization quality (LSE-D and LSE-C), and speed (FPS) are presented with standard deviations, quantifying the variability arising from factors including random noise initialization and stochastic sampling during inference.

In Table 8, we report more comparisons on the CelebV-Text (Yu et al., 2023) testset. Our model shows the best quality compared to other 3D model or video diffusion model based approaches.

# B   DETAILED ARCHITECTURE

Figure 9 illustrates the detailed architecture of the proposed model, comprising the visual encoder $E_v$, the decoder $D$, and the implicit motion generator $M$.

**Visual Encoder and Decoder**   The visual encoder $E_v$ is designed to disentangle appearance and motion from a given frame. It consists of multiple residual blocks followed by downsampling layers, projecting visual features into a motion vector $m$ and a set of appearance tensors $\{A_l \mid l = 1, \ldots, n_l\}$. The motion vector is further processed through a fully connected layer and $\tanh$ activation. These features are fed into the decoder $D$, which reconstructs the motion-transferred image via an upsampling path and a local flow module.

The decoder receives the motion vector $m$ and injects it into a latent axis alongside a $4 \times 4$ constant tensor, then processes it through the local flow module, which applies region-specific warping to the

features. This module is trained with a local flow loss to enforce spatial specificity. The decoder outputs the final RGB image using a ToRGB layer, and the full reconstruction is supervised via a reconstruction loss.

**Implicit Motion Generator**   The implicit motion generator $M$ follows a denoising diffusion paradigm, conditioned on speech and motion statistics. It takes as input a noised motion sequence $\{m_f^t\}$, a speech vector sequence $\{s_f\}$, and motion statistics: the mean $m^\mu$, the standard deviation $m^\sigma$, and a motion hint $m'$. These are projected via linear layers and modulated with positional encodings and diffusion time $t$.

The core of the generator consists of $L$ stacked transformer-style layers with self-attention and framewise temporal modulation. Framewise temporal guidance is implemented by incorporating speech-dependent conditioning at each layer. Motion statistics guidance is injected through residual connections to control expressiveness and pose stability across time. The final output is the predicted noise $\{\hat{\epsilon}_f\}$ for each frame, used in the denoising process to generate coherent motion aligned with the speech.

This modular architecture enables high-quality, real-time talking head generation without relying on explicit facial priors.

## C    LARGE LANGUAGE MODELS USAGE

We disclose the use of Large Language Models (LLMs) to enhance the writing and coding processes for this paper. Specifically, we used **Google Gemini**[2] to aid in refining and polishing the overall manuscript writing. Additionally, we utilized **Cursor**[3] and **Claude Code**[4] (or similar specific product page if available, using the main company link as a default) for generating and debugging scripts related to data preparation and experimental setup. The contributions of these models were limited to assistive tasks and did not rise to the level of a contributing author.

---

[2]https://gemini.google.com/
[3]https://www.cursor.so/
[4]https://www.anthropic.com/product/claude

