# OpenReview forum: "IF-MDM: Implicit Face Motion Diffusion Model for Compressing Dynamic Motion Latent"
_ICLR.cc/2026/Conference — ICLR 2026 Conference Withdrawn Submission_

### Official Review · Reviewer_pRXn · 2025-10-30

**Soundness:** 3
**Presentation:** 3
**Contribution:** 2
**Rating:** 2
**Confidence:** 4

**Summary:**

The paper proposes IF-MDM, a two-stage pipeline for audio-driven talking head generation: (1) a self-supervised encoder-decoder with  a Local Flow Module that disentangles visual appearance from motion dynamics and (2) a speech-conditioned diffusion model (UNet-like) that generates sequences of motion vectors using motion statistics guidance (mean/std) and a frame wise temporal guidance. The authors claim real-time 512x512 synthesis (at 45 fps) and generalization beyond humans (e.g., cats) with superior quality on HDTF and CelebV-Text dataset.

**Strengths:**

1. The paper is well written and clear.
2. Clear factorization of appearance and motion with an explicit local flow design sounds good and ablations show groupings matter. And motion mean and standard deviation provide practical controllability knobs, which makes the method more effective.
3. Extensive ablations validate the effect of each choice.

**Weaknesses:**

1. Evaluation is narrow
HDTF test set is only 30 videos; variance and identity leakage risks are not addressed and the test set is too small. It is unclear why such a small subset as chosen despite the availability of full/standard test sets.  On Celebv-Text they switch lip sync metrics from LSE-D/C to LSD-D/C (naming error?) , and VideoScore-TC, CSIM and FPS are not used here. This inconsistency breaks comparability and weakens conclusions. Standardize metrics, report identity-disjoint splits, and justify VideoScore-TC against human judgments.

2. User study missing.
The paper reports no user study or human preference evaluation. Automatic metrics alone are known to weakly capture perceived quality, faithfulness and controllability. Thus the claimed improvements remain insufficiently substantiated.

3. Metric-objective mismatch for ablations.
PSNR measures pixel wise reconstruction, not the paper's core goals (extracting the implicit motion, accurate lip-sync, temporal coherency..) In Table3 and Table4 where they checked the effects of G (number of group) and D (motion vector dimension), respectively, PSNR alone cannot reveal boundary artifacts, drift across chunks, occlusion failures, or identity leakage. Since G and D directly govern motion extraction capacity and stability, evaluating them with PSNR alone does not test the very behaviors that paper claims to improve.

**Questions:**

1. Why PSNR as the primary criterion for G (number of group) and D (motion vector dimension)?
Please justify its relevance to lip-sync, temporal stability and identity preservation and demonstrate that PSNR correlates with those metrics and is the criterion to choose the optimal ones. Re-run ablations 3 & 4 with the same metric suite used in the main tables (or other suitable metric?).

2. Clarification on conditioning
Your method injects temporal (speech-related) guidance into self-attention. Please justify this design choice rigorously against standard alternatives (cross-attention to audio tokens, FiLM/adapters, residual... )

3. Visualization needed
The paper claims that the proposed local flow module decomposes the global motion embedding into spatially localized flow fields, each designed to "attend to semantically meaningful regions such as the lips, eyes and torso". Could you please provide concrete evidence (e,g,, visualization) that each flow group consistently specializes to those regions rather than diffuse or identity-correlated patterns? Please report these alongside the main metrics under matched computed to demonstrate that the semantic alignment actually holds in practice.

---

### Official Review · Reviewer_JGc1 · 2025-10-30

**Soundness:** 2
**Presentation:** 2
**Contribution:** 1
**Rating:** 2
**Confidence:** 4

**Summary:**

This paper proposes IF-MDM, a fully self-supervised framework for audio-driven talking head generation. Unlike methods relying on explicit facial priors (e.g., 3DMM or landmarks), IF-MDM learns an implicit motion representation from video data, which helps avoid visual misalignment issues and improves the naturalness of full-frame video generation. The framework operates in two stages: (1) a local flow module disentangles appearance and fine-grained motion in a compressed latent space; (2) a diffusion-based motion generator produces temporally coherent motion sequences conditioned on speech, with motion statistics and framewise temporal guidance to enhance speech-motion alignment and controllability. Experiments on HDTF and CelebV-Text datasets show that IF-MDM achieves good performance in visual quality, temporal consistency, and lip-sync accuracy, while generating high-fidelity 512×512 videos at over 30 fps, enabling real-time inference. Additionally, the model’s generalizability to non-human subjects (e.g., animals) is demonstrated, owing to its independence from human-specific priors.

**Strengths:**

* High Efficiency and Real-Time Performance: IF-MDM achieves real-time generation (over 30 fps) of high-resolution (512x512) videos by using a compressed implicit motion latent space, outperforming conventional video diffusion models in speed while maintaining high visual quality.
* Elimination of Explicit Human Priors: By using a fully self-supervised approach with implicit motion, the framework avoids reliance on explicit facial priors like 3DMM or landmarks. This mitigates common issues such as "floating head" artifacts and misalignment between the head and background, leading to more natural full-frame results.

**Weaknesses:**

* Poor Qualitative Results: The video results shown on this work's project page exhibit significantly lower quality compared to current mainstream methods (e.g., Sonic[1], Kling-Avatar[2], VASA-1[3], etc.), with noticeable flaws in lip-sync accuracy and dental clarity. This contradicts the quantitative comparisons presented in Table 1.
* Insufficient Comparisons: This work claims in the final paragraph of the introduction that the proposed method surpasses state-of-the-art approaches, which is not accurate. Among all the compared methods, only Hallo3[4] was proposed in 2025, while the others date back to 2024. More advanced contemporary methods such as MultiTalk[5], OmniAvatar[6], FantasyTalking[7], HunyuanVideo-Avatar[8], and EchoMimicV3[9] are not included in the comparison. Furthermore, this work only provides quantitative comparisons with other methods and lacks qualitative comparisons.
* Questionable Effectiveness of the First-Stage Training: In fact, audio-driven generation within an implicit motion space has been extensively studied (e.g., VASA-1, JoyVASA[10], Playmate[11], Takin-ADA[12], DreamTalk[13], etc.). All these methods are built upon established image animation frameworks (e.g., MegaPortraits[14] for VASA-1, LivePortrait[15] for JoyVASA and Playmate, FaceVid2Vid[16] for Takin-ADA) and utilize a DiT network to integrate various conditioning signals into features extracted by the animation model. In contrast, this work opts to construct its own motion space in the first stage, which does not fundamentally differ from using open-source methods like LivePortrait. The first-stage training in this work is overly simplistic and lacks experimental validation to demonstrate its effectiveness. This may be a key reason for the subpar overall video quality, such as the generated talking head videos lacking expressive facial emotions and diverse head poses.

[1]Ji, Xiaozhong, et al. "Sonic: Shifting focus to global audio perception in portrait animation." Proceedings of the Computer Vision and Pattern Recognition Conference. 2025.

[2]Ding, Yikang, et al. "Kling-avatar: Grounding multimodal instructions for cascaded long-duration avatar animation synthesis." arXiv preprint arXiv:2509.09595 (2025).

[3]Xu, Sicheng, et al. "Vasa-1: Lifelike audio-driven talking faces generated in real time." Advances in Neural Information Processing Systems 37 (2024): 660-684.

[4]Cui, Jiahao, et al. "Hallo3: Highly dynamic and realistic portrait image animation with diffusion transformer networks." arXiv e-prints (2024): arXiv-2412.

[5]Kong, Zhe, et al. "Let Them Talk: Audio-Driven Multi-Person Conversational Video Generation." arXiv preprint arXiv:2505.22647 (2025).

[6]Gan, Qijun, et al. "OmniAvatar: Efficient Audio-Driven Avatar Video Generation with Adaptive Body Animation." arXiv preprint arXiv:2506.18866 (2025).

[7]Wang, Mengchao, et al. "Fantasytalking: Realistic talking portrait generation via coherent motion synthesis." arXiv preprint arXiv:2504.04842 (2025).

[8]Chen, Yi, et al. "HunyuanVideo-Avatar: High-Fidelity Audio-Driven Human Animation for Multiple Characters." arXiv preprint arXiv:2505.20156 (2025).

[9]Meng, Rang, et al. "Echomimicv3: 1.3 b parameters are all you need for unified multi-modal and multi-task human animation." arXiv preprint arXiv:2507.03905 (2025).

[10]Cao, Xuyang, et al. "JoyVASA: portrait and animal image animation with diffusion-based audio-driven facial dynamics and head motion generation." arXiv preprint arXiv:2411.09209 (2024).

[11]Ma, Xingpei, et al. "Playmate: Flexible Control of Portrait Animation via 3D-Implicit Space Guided Diffusion." Forty-second International Conference on Machine Learning.

[12]Lin, Bin, et al. "Takin-ADA: Emotion Controllable Audio-Driven Animation with Canonical and Landmark Loss Optimization." arXiv preprint arXiv:2410.14283 (2024).

[13]Ma, Yifeng, et al. "DreamTalk: When Emotional Talking Head Generation Meets Diffusion Probabilistic Models." arXiv preprint arXiv:2312.09767 (2023).

[14]Drobyshev, Nikita, et al. "Megaportraits: One-shot megapixel neural head avatars." Proceedings of the 30th ACM International Conference on Multimedia. 2022.

[15]Guo, Jianzhu, et al. "Liveportrait: Efficient portrait animation with stitching and retargeting control." arXiv preprint arXiv:2407.03168 (2024).

[16]Wang, Ting-Chun, Arun Mallya, and Ming-Yu Liu. "One-shot free-view neural talking-head synthesis for video conferencing." Proceedings of the IEEE/CVF conference on computer vision and pattern recognition. 2021.

**Questions:**

* How can the effectiveness of the first-stage training be demonstrated? Would it be better to directly use open-source methods such as LivePortrait[1] or FaceVid2Vid[2]?
* Could the experimental section be enriched? This should include both qualitative and quantitative comparisons with current mainstream state-of-the-art methods.

[1]Guo, Jianzhu, et al. "Liveportrait: Efficient portrait animation with stitching and retargeting control." arXiv preprint arXiv:2407.03168 (2024).

[2]Wang, Ting-Chun, Arun Mallya, and Ming-Yu Liu. "One-shot free-view neural talking-head synthesis for video conferencing." Proceedings of the IEEE/CVF conference on computer vision and pattern recognition. 2021.

---

### Official Review · Reviewer_HJJn · 2025-10-31

**Soundness:** 3
**Presentation:** 2
**Contribution:** 3
**Rating:** 4
**Confidence:** 2

**Summary:**

This paper proposes a diffusion-based talking-head generation framework that models implicit motion representations without relying on 3D priors. The method introduces motion statistics (mean and variance) for controllable motion intensity, a framewise temporal guidance mechanism to improve speech–motion alignment, and a local flow module for region-specific facial motion.

**Strengths:**

1. The idea of representing motion through global statistics is interesting and provides a simple yet interpretable way to control motion intensity.
2. The work provides a reasonable amount of ablation and qualitative analysis, showing awareness of how different components—such as motion statistics, local flow, and temporal guidance—affect the final performance

**Weaknesses:**

1. Some architectural components (e.g., group conv, sequential 1D softmax normalization) lack clear justification or supporting references. A brief explanation or comparison with alternative choices would help clarify the design choice.
2. The explanation of the residual addition for self-attention in the temporal guidance module is somewhat brief. Including a short equation or schematic showing how the guidance is applied within the attention computation would make the mechanism easier to follow.
3. The role of motion statistics remains underexplained. Even after reviewing the supplementary video, it is unclear what specific motion aspects each parameter controls, and in some cases, the resulting differences in expression seem to affect lip-sync quality.
4. The qualitative analysis of the local flow module feels limited. Additional qualitative comparisons or visualizations  would better highlight its effect on localized motion.

**Questions:**

1. What specific motion attributes do the motion statistics represent, and is there any quantitative or qualitative analysis showing how adjusting them relates to measurable motion factors (e.g., pose variance, articulation strength) or affects lip-sync accuracy and stability?
2. Could the authors include more qualitative examples or visualizations to illustrate how the local flow module contributes to improved regional motion fidelity?

---

### Official Review · Reviewer_v3N2 · 2025-11-10

**Soundness:** 2
**Presentation:** 2
**Contribution:** 2
**Rating:** 2
**Confidence:** 4

**Summary:**

The paper proposes a self-supervised diffusion framework for audio-driven talking heads. It claims to (i) replace explicit head-motion priors with “implicit motion templates,” (ii) improve speech–motion alignment via a local flow module, motion-statistics guidance, and frame-wise temporal guidance, and (iii) deliver high-fidelity 512×512 videos at up to 45 FPS, capturing eye blinks/torso motion and generalizing to varied identities and even animals.

**Strengths:**

* Ambition to remove handcrafted priors and expert models and train end-to-end with SSL is timely.
* The three alignment aids (local flow, motion stats, framewise guidance) are reasonable inductive biases for stabilizing motion generation.
* Claimed runtime/resolution targets are attractive for practical systems.

**Weaknesses:**

* Undefined core concepts. “Implicit motion templates” and “motion-statistics guidance” are central but not clearly defined or formalized; their computation, role at train/inference, and why they solve SSL spurious correlations are unclear.
* Motivation for the latent space and local flow module. The paper asserts a special, LF-module–based latent space “tailored for dynamic motion,” but offers no concrete evidence for why it is special or better than standard latent formulations. The impact of the local flow module is not demonstrated in the supplemental videos.
* Progressive data structuring. The introduction claims multi-scale improvements from “progressive levels,” but these levels, training curriculum, and measured gains are not specified or ablated.
* Dataset filtering. Splitting data into face-only, head-only, and face+head sets is plausible, yet the paper does not quantify how much this filtering helps stability/quality or report failure modes.
* Latent-based model “computational challenges.” This is mentioned but not substantiated with memory/throughput numbers, complexity analysis, or comparisons; claims remain anecdotal.
* Overstated generalization. Claims of robust eye-blink/torso capture and generalization to animals require qualitative evidence (videos) and task-specific metrics; these are missing or insufficient.
* Comparative evaluation gaps. Important recent baselines (InfiniteTalk, OmniAvatar/OmniSync, Hallo3, FLOAT, Generative Motion Latent FM for Talking Portrait) are absent from qualitative comparisons, limiting the strength of the results.
* he effectiveness of individual components (local flow, motion stats, framewise guidance) is not clearly shown in video; without this, it is hard to assess the real contribution.

**Questions:**

1. What computational challenges of prior latent models does IF-MDM avoid? Please give concrete memory/latency numbers and ablate vs. a strong latent baseline.
2. What is “motion-statistics guidance”? Specify how m^\mu, m^\sigma (or equivalents) are computed (per-sequence? global?), how they are used at inference, and why they mitigate SSL spurious correlations.
3. Clarify L049 (“eliminating reliance on human expert models for extracting motion liveness”). Which expert models are avoided, and what replaces them in your pipeline?
4. Show component impact. Provide side-by-side videos and metrics isolating (a) local flow, (b) motion-stats guidance, (c) framewise temporal guidance; quantify each gain. Also add qualitative comparisons vs. InfiniteTalk, OmniAvatar/OmniSync, Hallo3, FLOAT.

---

### Note · Authors · 2025-11-13

**Comment:**

After reviewing and considering the reviewers’ comments, we have decided to further strengthen and refine the content in the next revision. We appreciate the reviewers’ efforts and thoughtful feedback.

**Withdrawal Confirmation:**

I have read and agree with the venue's withdrawal policy on behalf of myself and my co-authors.